# ESCAPING FLAT AREAS VIA FUNCTION-PRESERVING STRUCTURAL NETWORK MODIFICATIONS

## ABSTRACT

Hierarchically embedding smaller networks in larger networks, e.g. by increasing the number of hidden units, has been studied since the 1990s. The main interest was in understanding possible redundancies in the parameterization, as well as in studying how such embeddings affect critical points. We take these results as a point of departure to devise a novel strategy for escaping from flat regions of the error surface and to address the slow-down of gradient-based methods experienced in plateaus of saddle points. The idea is to expand the dimensionality of a network in a way that guarantees the existence of new escape directions. We call this operation the opening of a tunnel. One may then continue with the larger network either temporarily, i.e. closing the tunnel later, or permanently, i.e. iteratively growing the network, whenever needed. We develop our method for fully-connected as well as convolutional layers. Moreover, we present a practical version of our algorithm that requires no network structure modification and can be deployed as plug-and-play into any current deep learning framework. Experimentally, our method shows significant speed-ups.

## 1 INTRODUCTION

Training deep neural networks involves the minimization of a non-convex, (piecewise) smooth error function defined over a high-dimensional space of parameters. Such objectives are most often optimized by stochastic gradient descent or variants thereof Duchi et al. (2011); Zeiler (2012); Kingma & Ba (2014). One of the main difficulties occurring in this minimization problem is caused by the proliferation of saddle points, often surrounded by flat areas, which may substantially slow down learning Dauphin et al. (2014). Escaping plateaus around saddle points is of critical importance not only to accelerate learning, but also because they can be mistaken for local minima, leading to poor solutions if learning were to be stopped too early. Recent approaches to analyze and deal with this challenge are often based on adding noise, e.g. Jin et al. (2017).

Here, we pursue a different philosophy. Inspired by the seminal paper of Fukumizu & Amari (2000), we propose to expand a network by creating an additional hidden unit or filter, whenever optimization is slowing down too much. This unit is obtained by a simple cloning process of a feature or feature map, combined with a fast to execute optimization that scales outbound weights. The latter explicitly aims at opening-up an escape direction by maximizing the gradient norm in the expanded network. In particular, we exploit this ability to devise a method to escape saddle points, bad local minima and general flat areas in neural network optimization, which we call *Tunnels*. After taking a small number of update steps in the expanded network, we can decide to project back to the original network architecture. The hope is that the flat area or bad local minimum will have been left behind.

Our contributions are as follows:

- We extend the network embedding idea of Fukumizu & Amari (2000) to deep networks and CNNs.

- We show that in contrast to the case of a three-layer perceptron, the newly constructed network has a non-zero gradient, even if the original network did not.

- We derive a closed-form solution for cloning a unit or feature map to maximize the norm of this gradient.

- We develop a *practical version* of our algorithm that requires no network structure modification and can be deployed as plug-and-play into any current deep learning framework.
- We show experimentally that our method escapes effectively from flat areas during training.

In Section 2, we provide background, motivation, and explain in detail how to *open the tunnel* by solving an interposed optimization problem. We discuss the fully connected case as well as the practically relevant case of convolutional layers. In Section 3, we describe how to *close the tunnel*, *i.e.* how to project back the expanded network onto one of the original dimensionality. We also propose a practical version of our idea that does not require an explicit modification of the network structure, based on a well-chosen reorganization of the weights. Finally, experiments and related work are presented in Sections 4 and 5 respectively.

## 2    TUNNEL OPENING

### 2.1    REDUCIBLE NETWORKS

Feedforward neural networks can be defined in terms of a directed acylic graph $(U, E)$ of computational units. Concretely, we assume that each unit $u \in U$ is equiped with an activation function $\sigma_u$ and a bias parameter $b_u \in \mathbb{R}$. Moreover, each edge $(u, v) \in E$ has an associated weight $w_{uv} \in \mathbb{R}$, where we implicitly set $w_{uv} = 0$ for $(u, v) \notin E$.

We can interpret a network computationally by assigning values $x_u$ to source or input units and propagating these values through the DAG using local computations with ridge function $x_v = \sigma_v(b_v + \sum_u w_{uv} x_u)$ all the way to the sink or output units. Ignoring values at intermediate, hidden nodes, a network thus realizes a transfer function $F(\cdot; \{w, b\}) : \mathbb{R}^n \to \mathbb{R}^m$, where $n$ and $m$ denote the number of input and output nodes, respectively. We call two networks *input-output equivalent*, if they define the same transfer function.

It is a fundamental question whether the parameterization $\{w, b\} \mapsto F(\cdot; \{w, b\})$ is one-to-one and, if not, what the sources of non-uniqueness are. As investigated in the classical work of Sussmann (1992), one source of non-uniqueness can be tied to specific properties of the parameters, which make the network *reducible* to a smaller network. There are precisely three such conditions on hidden units, which do not depend on special properties of the activation function[1]:

$$\text{Inconsequential unit } u' \quad \forall v : w_{u'v} = 0 \tag{1a}$$

$$\text{Constant unit } u' \quad x_{u'} = \text{const} \tag{1b}$$

$$\text{Redundant units } u', u \quad x_{u'} = \pm x_u \tag{1c}$$

(1a) says that $x_{u'}$ is ignored downstream as all outbound weights are zero. Note that this means, that the inbound weights can be chosen arbitrarily, in particular we may transform $w_{u'v} \mapsto 0 \ (\forall v)$, thus converting $u'$ into an *isolated* node, without altering the network's transfer function. (1b) implies that each child $v$ of $u'$ can effectively be disconnected from $u'$ via the parameter transformation $(w_{u'v}, b_v) \mapsto (0, b_v + w_{u'v} x_{u'})$, rendering $u'$ inconsequential. Finally, (1c) permits the invariant parameter transformation $(w_{u'v}, w_{uv}) \mapsto (0, w_{uv} \pm w_{u'v})$, which reconnects all children of $u'$ with $u$ instead, again rendering $u'$ inconsequential.

A network is *reducible*, if at least one of its hidden nodes is reducible, otherwise it is *irreducible*. As shown rigorously in Sussmann (1992) for three-layer perceptrons with a single output, irreducible networks are minimal, *i.e.* they are not input-output equivalent to a network with fewer hidden units, and the remaining unidentifiability can be tied to a simple symmetry group (e.g. permutations and sign changes of hidden units).

### 2.2    NETWORK EXPANSION

We are interested in embedding smaller networks into larger ones by adding a single hidden node, such that the transfer functions remains invariant. If the smaller network is irreducible, then by the above result, the new unit will have to be reducible. This tells us that such extensions can only be obtained through reversing the operations in (1).

---

[1]Further redundancies may be introduced by specifics of $\sigma_u$.

Table 1: Parameter transformations for network extension.

Table 2: Consequences of network extension on gradients.

| CASE | | | SMALL | EXTENDED NET |
|------|---|---|-------|--------------|
| (A) | $\forall p$ | | $w_{pu'} = 0$ | $w_{pu'} \in \mathbb{R}$ |
| | $\forall v$ | | $w_{u'v} = 0$ | $w_{u'v} = 0$ |
| (B) | $\forall p$ | | $w_{pu'} = 0$ | $w_{pu'} = 0$ |
| | $\forall v$ | | $w_{u'v} = 0$ | $w_{u'v} \in \mathbb{R}$ |
| (C) | $\forall p$ | | $w_{pu} \in \mathbb{R}$ $w_{pu'} = 0$ | $w_{pu} \mapsto w_{pu}$ $w_{pu'} \mapsto w_{pu}$ |
| | $\forall v$ | | $w_{uv} \in \mathbb{R}$ $w_{u'v} = 0$ | $w_{uv} \mapsto \lambda_v w_{uv}$ $w_{u'v} \mapsto (1 - \lambda_v) w_{uv}$ |

| CASE | SMALL | EXTENDED NET |
|------|-------|--------------|
| (I) | $x_{u'} = 0$ | $x_{u'} \in \mathbb{R}$, $\dfrac{\partial \mathcal{E}}{\partial w_{u'v}} \in \mathbb{R}$ $(\forall v)$ |
| (II) | $\delta_{u'} = 0$ | $\delta_{u'} \in \mathbb{R}$, $\dfrac{\partial \mathcal{E}}{\partial w_{pu'}} \in \mathbb{R}$ $(\forall p)$ |

Assume that we have an error function $\mathcal{E} : \mathbb{R}^m \to \mathbb{R}$, inducing a risk $\mathcal{R}(F) = \mathbf{E}\left[\mathcal{E}(F(\mathbf{x}))\right]$, defined empirically in practice. We consider the set of critical points of a network via the condition $\nabla_{\{w,b\}} \mathcal{R}(F(\cdot; \{w, b\})) = 0$. Fukumizu & Amari (2000) have investigated how the critical points of a network change after adding a reducible unit, one of the main results being that a global minimum of the smaller network can induce affine subspaces of local minima and saddle points in the extended network. This analysis only holds for the special case of a single downstream (*i.e.* output) unit. In general, embedding a smaller network in an input-output equivalent larger one may introduce directions of non-vanishing error derivatives, even at critical points. Critical points of a smaller network are not guaranteed to remain critical points after extension. This is the motivation of why one can potentially create escape tunnels from regions near poor critical points by such operations.

Specifically, we have the options to add a node $u'$: (A) with non-zero outbound weights, or (B) with non-zero inbound weights, or (C) by cloning it from an existing unit $u$ and re-weighting its outbound weights. Formally, this can be realized by parameter transformations, as described[2] in Table 1.

Let us first state what can be said about the partial derivatives, after a new unit has been added. In a DAG one can compute $\partial \mathcal{E} / \partial w_{u'v}$ for fixed inputs in three steps: *(i)* computing all activations $x_p$ via forward propagation *(ii)* computing deltas $\delta_p \equiv \partial \mathcal{E} / \partial x_p$ via backpropagation from sink nodes to source nodes and *(iii)* in a local application of the chain rule

$$\frac{\partial \mathcal{E}}{\partial w_{u'v}} = \delta_v \, x'_v \, x_{u'} \,. \tag{2}$$

This results in the situation described in Table 2 for the activation of inbound (I) or outbound (II) weights.

Intuitively speaking, by activating the outbound weights, the unit is no longer inconsequential and the inbound weights matter, whereas by activating the input weights, what was a constant (and thus useless) node before, turns into a selective and potentially useful feature for downstream computations. It is not clear how to best initialize the new weights. One option is to proceed in the spirit of methods such as Breiman (1993) and more broadly boosting algorithms Friedman et al. (2000); Buehlmann (2006), which iteratively fit a residual loss, clamping all or some of the existing parameters. However, this seems (i) computationally unattractive on present-day computers, and (ii) pursues the philosophy of incremental learning, whereas our objective is not to greedily grow the network unit by unit, but rather to temporarily open an escape tunnel.

## 2.3 NODE CLONING

First, we look at a motivational example of an existing node $u$ with two children nodes $v_1, v_2$. The backpropagation formula yields $\delta_u = \delta_u^1 + \delta_u^2$, where $\delta_u^i = \delta_{v_i} w_{uv_i} x'_{v_i}$, $i \in \{1, 2\}$. This splits the effect that $u$ has on the error via the computational paths leading through $v_1$ and $v_2$, respectively. The partial derivative of an inbound weight from some parent $p$ of $u$ in the original network is then simply

$$\frac{\partial \mathcal{E}}{\partial w_{pu}} = \left(\delta_u^1 + \delta_u^2\right) x'_u \, x_p \,. \tag{3}$$

---

[2] $\alpha \in \mathbb{R}$ expresses the fact that a parameter is unconstrained.

Let us define the shortcuts $a_{pi} := \mathbf{E}[\delta_u^i x_u' x_p]$. At a critical point it holds that $a_{p1} + a_{p2} = 0$, *i.e.* $a_p := a_{p1} = -a_{p2}$. If we clone $u$ into $u'$ (case (C) in Table 1), then

$$\mathbf{E}\left[\frac{\partial \mathcal{E}}{\partial w_{pu}}\right] = (\lambda_1 - \lambda_2)a_p = -\mathbf{E}\left[\frac{\partial \mathcal{E}}{\partial w_{pu'}}\right] \tag{4}$$

In particular, we can see that by choosing $\lambda_1 = \lambda_2$ we remain at a critical point and that the contributions to the squared gradient norm are maximal for $\lambda_1 = 1 - \lambda_2 \in \{0, 1\}$. This holds irrespective of the value of $a_w$ and hence irrespective of the chosen upstream unit $p$. Note that we could consider increasing weights even further by choosing $\lambda_1 > 1$ and $\lambda_2 < 0$ (or vice versa). However this would artificially increase the gradient norm by making outbound weights of $u$ and $u'$ arbitrarily large.

In the general case, where $u$ has in-degree $K$ and out-degree $L$, we can use the same quantities $a_{pi}$ as above, which we can summarize in a matrix $A = (a_{pi}) \in \mathbb{R}^{K \times L}$. We can now define an objective quantifying the contribution of the two modified units $u$ and $u'$ to the squared gradient norm obtained after cloning:

$$\mathcal{H}(\lambda) = \sum_p \mathbf{E}\left[\frac{\partial \mathcal{E}}{\partial w_{pu}}\right]^2 + \mathbf{E}\left[\frac{\partial \mathcal{E}}{\partial w_{pu'}}\right]^2 \tag{5}$$

which can also be written as

$$\mathcal{H}(\lambda) = \lambda^T A^T A \lambda + (\mathbf{1} - \lambda)^T A^T A (\mathbf{1} - \lambda). \tag{6}$$

Note that at a critical point, we have that the columns of $A$ add up to the zero vector $\sum_i a_{\cdot i} = \mathbf{0}$, in which case the above quantity simplifies as $\mathcal{H}(\lambda) = 2\lambda^T A^T A \lambda$.

At a critical point, this is a convex maximization problem which achieves its minimal value of 0 at any $\lambda \in \mathbb{R}\mathbf{1}$. Maximizing $\mathcal{H}$ over the unit hyper-cube will in many cases degenerate to an integer problem, which may not be fast to solve. Also, we generally prefer to not make too drastic changes to the network, when opening the tunnel. Hence, we pursue the strategy of taking some $\lambda^0 \in \mathbb{R}\mathbf{1}$ as a natural starting point and considering $\lambda^t = \lambda^0 + t\lambda^*$, where $\lambda^*$ is a unit length principal eigenvector of $A^\top A$. This is equivalent to maximizing $\mathcal{H}$ over the sphere of radius $t$, centered at $\lambda^0$. We choose $\lambda^0 = \frac{1}{2}\mathbf{1}$ as it minimizes $\mathcal{H}$ over $\mathbb{R}\mathbf{1}$ in the general case, *i.e.* at a non-critical point. Note that $\mathcal{H}(\lambda^t) = (t\rho)^2$, where $\rho$ is the largest eigenvalue.

The tunnel opening we propose in order to escape saddles therefore consists in cloning an existing unit $u$ into a new unit $u'$, while assigning them the outbound weights of $u$ respectively rescaled by $\lambda^t$ and $\mathbf{1} - \lambda^t$, where $t > 0$ is a hyperparameter controlling the norm of the newly obtained gradient.

## 2.4 Filter cloning

A particular instance of neural networks used in practice includes convolutional neural networks. Since such networks may be very deep and are often parametrized by millions of parameters, their optimization can be challenging. In order to implement our proposed method for these networks, one needs to adapt the convex optimization problem of Eq. (6) and find the correponding matrix $A$.

We consider here cloning a filter of a convolutional layer followed by a non-linearity and either a convolutional or fully-connected layer. We derive computations in the first case, the second being similar and presented in appendix A.

Convolution between single-channel image $x$ and filter $K$ is defined by

$$[x \star K]_{i,j} = \sum_{r,s} [K]_{rs}[x]_{r+i,s+j}, \tag{7}$$

where $[\cdot]_{ij}$ denotes the pixel at the $i^{th}$ row and $j^{th}$ column.

A convolutional layer with pointwise non-linearity computes the following activation:

$$[x_v]_{ij} = \sigma_v(b_v + \sum_u [x_u \star K_{uv}]_{ij}), \tag{8}$$

where $u$ is the channel index. By simple application of the chain rule, we have

$$\frac{\partial \mathcal{E}}{\partial [K_{pu}]_{ij}} = \sum_{v,r,s} \frac{\partial \mathcal{E}}{\partial [x_v]_{rs}} \frac{\partial [x_v]_{rs}}{\partial [K_{pu}]_{ij}}. \tag{9}$$

By a well-chosen chain rule we now have

$$\frac{\partial [x_v]_{rs}}{\partial [K_{pu}]_{ij}} = \sum_{i',j'} \frac{\partial [x_v]_{rs}}{\partial [x_u]_{i'+r,j'+s}} \frac{\partial [x_u]_{i'+r,j'+s}}{\partial [K_{pu}]_{ij}} = \sum_{i',j'} [K_{uv}]_{i'j'} [x'_v]_{rs} \frac{\partial [x_u]_{i'+r,j'+s}}{\partial [K_{pu}]_{ij}}, \tag{10}$$

Like before, let us define the shortcuts $[\delta_v]_{rs} := \partial \mathcal{E} / \partial [x_v]_{rs}$, and

$$a_{(pij),(vi'j')} := \mathbf{E}\left[ [K_{uv}]_{i'j'} \sum_{r,s} [x'_v]_{rs} [\delta_v]_{rs} \frac{\partial [x_u]_{i'+r,j'+s}}{\partial [K_{pu}]_{ij}} \right]. \tag{11}$$

Let us summarize these in a matrix $A = (a_{(pij),(vi'j')})$ where $(pij)$ indexes a row and $(vi'j')$ a column. Now, cloning a filter $u$ into $u'$ would lead to the substitutions $[K_{uv}]_{i'j'} \leftarrow \lambda_{(vi'j')}[K_{uv}]_{i'j'}$ and $[K_{u'v}]_{i'j'} \leftarrow (1 - \lambda_{(vi'j')})[K_{uv}]_{i'j'}$ for some scaling vector $\lambda$, yielding:

$$\mathbf{E}\left[ \frac{\partial \mathcal{E}}{\partial [K_{pu}]_{ij}} \right] = (A\lambda)_{pij} \quad \text{and} \quad \mathbf{E}\left[ \frac{\partial \mathcal{E}}{\partial [K_{pu'}]_{ij}} \right] = (A(\mathbf{1} - \lambda))_{pij}. \tag{12}$$

The contribution of the two modified filters $u$ and $u'$ to the squared gradient norm obtained after cloning is again given by

$$\mathcal{H}(\lambda) = \lambda^T A^T A \lambda + (\mathbf{1} - \lambda)^T A^T A (\mathbf{1} - \lambda), \tag{13}$$

and the same analysis holds.

## 3 TUNNEL CLOSING

### 3.1 AVERAGING INBOUND, SUMMING OUTBOUND WEIGHTS

One can simply go back to a network of the original size by deleting the newly added unit (resp. filter) $u'$, replacing the inbound weights $w_{pu}$ of the previously cloned unit $u$ by the average $(w_{pu} + w_{pu'})/2$ and its outbound weights $w_{uv}$ by the sum $w_{uv} + w_{u'v}$. Note that closing the tunnel in this way just after opening it will leave the original network unchanged, but we aim to perform a few gradient steps between opening and closing, in which case this heuristic does not guarantee function preservation.

### 3.2 PRACTICAL CLOSING-OPENING VIA WEIGHT REORGANIZATION

Instead of opening the tunnel by adding a new unit (resp. filter), and then closing it later, we propose a weight reorganization bypassing the network structure modification. The idea is the following: select two units $u_1$ and $u_2$ for closing, and a unit $u$ to clone. Start by closing $u_2$ on $u_1$ by averaging:

$$w_{pu_1} \mapsto (w_{pu_1} + w_{pu_2})/2, \quad w_{u_1 v} \mapsto w_{u_1 v} + w_{u_2 v} \tag{14}$$

and then open by cloning $u$ using the spot 'left free' by $u_2$:

$$w_{pu_2} \mapsto w_{pu}, \quad w_{u_2 v} \mapsto (1 - \lambda_v^t)w_{uv}, \quad w_{uv} \mapsto \lambda_v^t w_{uv} \tag{15}$$

for all $v, p$.

This method has the benefit of requiring no structural modification, making tunnels easier to use in practice.

## 4 EXPERIMENTS

In the following experiments, we train a variety of CNNs on standard image classification datasets. We then test variations of our method for escaping plateaus in optimization. In stochastic settings, we repeat all experiments ten times and plot mean and one standard deviation over the repetitions. Detailed experimental setup information as well as full length runs can be found in the Appendix.

### 4.1 ADDING FILTERS TO A SINGLE HIDDEN CONVOLUTIONAL LAYER

In order to demonstrate the effectiveness of our method for escaping plateaus, we start out on a small toy CNN containing a single hidden convolutional layer with 8 filters and a sigmoid nonlinearity, followed by a fully connected softmax layer. The network is trained on the MNIST dataset. Due to the small size of our network, we are able to employ *Newton's method* to reliably find flat areas in the optimization landscape. Once a flat area has been found, we use SGD to train the network for the rest of the training.

At step 50, we perform tunnel opening and add 2 filters to the hidden layer. The same 2 filters are removed again when we perform tunnel closing at step 300. To stay in accordance to our theory, we restrict ourselves to the most basic building blocks of neural networks. This means that we use a straightforward 2D convolution, followed by a sigmoid nonlinearity. Notably, we do not employ tricks such as batch normalization, pooling or residual connections.

Results are shown in Figure 1. Note the immediate drop in loss when we add the filters. Even after we close the tunnel and remove the filters again, the model continues to remain unstuck, therefore we conclude that we have successfully escaped the flat area. Also note the comparison between adding filters using the proposed $\lambda^t$ versus adding filters in a random fashion (using a randomly sampled $\lambda$ with equal magnitude as $\lambda^t$), which provides no improvement. Further, we observe empirically that we can find and get stuck in flat areas with equal ease even when using networks of initially higher capacity, *i.e.* as if the tunnel had been open from the beginning of training. This suggests that the occurrence of flat areas is not merely a problem of *network capacity* and that the targeted manipulation of network structure can be used to escape such flat areas with success.

### 4.2 MEASURING THE IMPACT OF $\lambda^t$

Having developed a theory that includes as a hyperparameter the choice of $t$ to control the norm of the gradient after tunnel opening, we investigated the influence of this parameter in a setting that allows us to eliminate stochasticity by computing gradients over the entire dataset. We used full batch gradient descent to optimize the setup from Section 4.1 to convergence (negligible gradient norm, no change in loss). We then applied tunnel opening using different choices of $t$ to compute $\lambda^t$ and measured the resulting norm of the full batch gradient.

Results can be seen in Figure 1. Clearly, the choice of $t$ has a direct influence on the resulting gradient norm. Note the quadratic relationship between $t$ and the gradient norm as well as the additive effect of the number of filters added, both as predicted by our theory. We also measured the total loss and could confirm that it remains unchanged after tunnel opening, irrespective of the choice of $t$, again as our theory predicts.

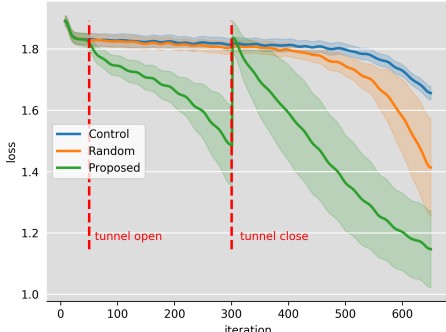
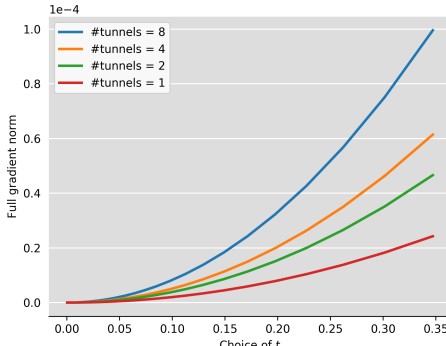

Figure 1: **Left:** Training loss of a single-hidden-layer CNN on MNIST. Blue: Original network stuck in a flat area. Green: We add two filters at step 50 using our tunnel opening technique and we remove them again at step 300. Orange: We add and remove the same two filters, but instead of our gradient maximization technique, we add the filters in a random fashion. **Right:** Resulting full gradient norm after tunnel opening in a fully optimized single-hidden-layer CNN on MNIST against the choice of hyperparameter $t$. Each curve refers to a different number of tunnels opened (i.e. filters added).

### 4.3 PRACTICAL WEIGHT REORGANIZATION FOR A SINGLE HIDDEN CONVOLUTIONAL LAYER

Figure 2 shows the practical closing-opening version of our algorithm presented in Section 3.2, with the same initial setup as in Section 4.1. As before, we use Newton's method to find an appropriate flat area and run SGD afterwards. At step 50, we perform weight reorganization on 2 filters in the convolutional layer, which means we first use tunnel closing to merge 4 filters into 2, then we perform tunnel opening to add back the 2 filters we lost in the process. We compare to two baselines: Perturbed gradient descent (Jin et al., 2017) and NEON (Xu & Yang, 2017), both also after step 50.

As can be seen, the targeted weight reorganization method not only escapes from the flat area, but also vastly outperforms both baselines as well as random weight reorganization (again, of equal magnitude) despite not changing the network architecture, but merely modifying already existing weights. This is further evidence that the success of our method can be attributed to the particular form of weight manipulation and not simply to an increase in network capacity.

### 4.4 TUNNEL OPENING IN A DEEP NEURAL NETWORK

Lastly, we want to show that it is also possible to apply tunnel opening to just part of a larger and deeper network and still benefit its optimization. In addition, we would like to demonstrate the applicability of our method in more realistic settings, thus we drop a few of our initial restrictions. We train a CNN with 5 hidden convolutional layers of 64 filters each on the ImageNet dataset, using max pooling, batch normalization and ReLU nonlinearities. Note that we apply tunnel opening only to the last two convolutional layers, such that batch normalization and pooling, though applied in the network, are not interfering with our derived closed-form solution.

Most importantly, we no longer use Newton's method in order to find a saddle point. Instead, we restart optimization from different random initializations until we find an initialization that robustly gets stuck in a flat area. Further, to make our experiments closer to practical use, we no longer use plain SGD, but train the network using Adam Kingma & Ba (2014). We perform tunnel opening at step 5000, where we add a single filter to each of the last two hidden layers. After opening, we clear the Adam and BatchNorm accumulation buffers and decrease the learning rate for the first 20 steps in order to minimize unwanted warm-restart effects (this is equally done in the control runs).

Results are displayed in Figure 2. As can be seen, the random tunnel opening does not manage to escape the plateau, whereas the proposed method does so successfully and faster than the baseline methods. This shows that the method can be applied even in individual parts of deep networks in order to benefit the optimization of the entire network.

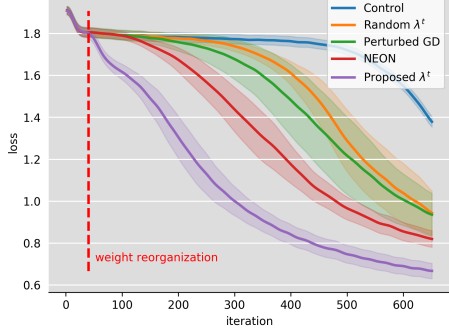 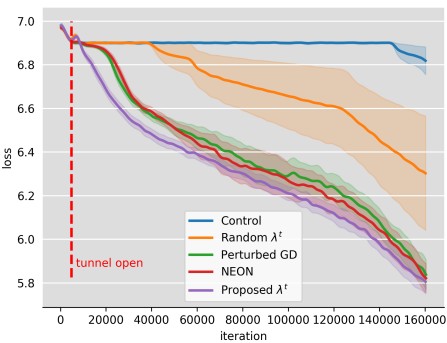

Figure 2: **Left** Training loss of a single-hidden-layer CNN on MNIST. Blue: Original Network stuck in a flat area. Orange: Random weight reorganization at step 50. Purple: Proposed weight reorganization at step 50. Shaded areas represent one standard deviation around the mean of 10 repeated experiments. **Right:** Training loss of a five-layer CNN on ImageNet. Blue: Original Network stuck in a flat area. Orange: Random tunnel opening at step 5000. Purple: Proposed tunnel opening at step 5000. Shaded areas represent one standard deviation around the mean of 10 repeated experiments. **Baselines:** Green: Perturbed GD (Jin et al., 2017). Red: NEON (Xu & Yang, 2017).

## 5 COMPARISON TO RELATED WORK

Our ideas are inspired by the work of Fukumizu & Amari (2000), which proposes different ways to embed a single-hidden-layer neural network into a bigger one while preserving its output values at any point. They further prove that such networks can always be added a hidden unit such that each critical point of the smaller network would induce a myriad of critical points in the augmented network. This increase in the quantity of critical points for bigger networks was rigorously analyzed in the specific case of linear models with quadratic loss in Kawaguchi (2016), where the author proves in particular that linear models are more prone to possessing degenerate saddle points with depth, and generalizes his study to deep non-linear models under some independence assumption; Dauphin et al. (2014) argue that such a proliferation of saddle points, often surrounded by flat areas, also called plateaus, constitutes one of the main obstacles to training deep neural networks with gradient-based algorithms, because of the incurred slow-down.

Different solutions have been proposed in order to cope with the saddle-point problem. In the case of strict saddles, *i.e.* critical points where the Hessian has at least one negative eigenvalue, it has been shown that the presence of noise in SGD would enable it to escape from these strict saddles in a polynomial number of iterations Ge et al. (2015), while injecting some supplementary noise could improve the escaping rate Jin et al. (2017); Xu & Yang (2017). Second order methods were also designed specifically to escape from saddles Dauphin et al. (2014); Reddi et al. (2017); globalized (or regularized) Newton methods such as Trust Region Conn et al. (2000) and cubic regularizations Nesterov & Polyak (2006), as well as recent stochastic variants Kohler & Lucchi (2017), escape strict saddles by computing second-order information, and Anandkumar & Ge (2016) propose a method using third order derivatives to escape from degenerate saddles that are indistinguishable from local minima with their first and second order derivatives. Lastly, Bengio et al. (2006); Bach (2017) show that the optimization of neural networks can be posed as a convex optimization problem if the number of hidden units is variable, thereby avoiding the problem entirely.

Although escaping from strict saddles seems very important, Sankar & Balasubramanian (2017) show that gradient-based algorithms on deep networks tend to converge towards saddles with high degeneracy. On the other hand, Chaudhari & Soatto (2017) argue that SGD run on deep neural networks converges to limit cycles where the loss is constant but the gradient norm is non-zero.

Previous work has also investigated changing the neural network's architecture while preserving its functional mapping: Chen et al. (2015); Cai et al. (2018); Lu et al. (2018) provide methods for adding hidden units or layers to trained neural networks. However, this is done in the context of transfer learning and architecture search and none provides explicit control over the resulting gradient of the expanded network.

To the best of our knowledge, our approach significantly differs from all existing work in the literature, in that without adding any supplementary noise or making use of higher order information, we propose to escape saddles by increasing the size of the network in a way that provenly gives a non-zero gradient, while preserving the network's output for any input. Further notice that our method has the 'a priori' potential to escape from bad local minima as well, since a tunnel can transport the network's weights to a completely different position in parameter space.

## CONCLUSION AND FUTURE WORK

Flat areas surrounding saddle points are believed to be a major problem in the optimization of deep networks via gradient-based algorithms, as they can slow-down learning or be mistaken for local minima. Developing methods to escape these saddles is therefore of crucial importance. We propose a new and original technique to escape plateaus, that can either be used by adaptively modifying the size of the network during training, or simply by reorganizing its weights if we are to avoid structural modifications. We provided theoretical guarantees of creating a new maximal gradient-descent direction *while preserving the loss* by opening a tunnel, and empirical evidence that both the structural modification and the weight reorganization versions of tunnels can efficiently escape saddles during training. Potential future work includes extending the technique to allow for batch-normalization and pooling operators following the layer on which tunneling is performed.

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

## A    FILTER CLONING FOLLOWED BY A FULLY-CONNECTED LAYER

We consider here cloning a filter of a convolutional layer followed by a non-linearity and then a fully-connected layer. Derivations are similar as in Section 2.4.

A fully connected layer with pointwise non-linearity, applied just after a convolutional layer $([x_u]_{ab})_{u,a,b}$ (where $u$ denotes the channel and $a, b$ the pixels) with pointwise non-linearity, computes the following scalar activation:

$$x_v = \sigma_v(b_v + \sum_{u,a,b} [x_u]_{ab}[w_{uv}]_{ab}). \tag{16}$$

By simple application of the chain rule, we have

$$\frac{\partial \mathcal{E}}{\partial [K_{pu}]_{ij}} = \sum_v \frac{\partial \mathcal{E}}{\partial x_v} \frac{\partial x_v}{\partial [K_{pu}]_{ij}}. \tag{17}$$

By a chain rule again, we obtain

$$\frac{\partial x_v}{\partial [K_{pu}]_{ij}} = \sum_{i',j'} \frac{\partial x_v}{\partial [x_{u'}]_{i',j'}} \frac{\partial [x_u]_{i',j'}}{\partial [K_{pu}]_{ij}} = \sum_{i',j'} [w_{uv}]_{i'j'} x'_v \frac{\partial [x_u]_{i',j'}}{\partial [K_{pu}]_{ij}}. \tag{18}$$

Like before, let us define the shortcuts $\delta_v := \partial \mathcal{E}/\partial x_v$, and

$$a_{(pij),(vi'j')} := \mathbf{E}\left[ [w_{uv}]_{i'j'} x'_v \delta_v \frac{\partial [x_u]_{i',j'}}{\partial [K_{pu}]_{ij}} \right]. \tag{19}$$

Let us summarize these in a matrix $A = (a_{(pij),(vi'j')})$ where $(pij)$ indexes a row and $(vi'j')$ a column. Now, cloning a filter $u$ into $u'$ would lead to the substitutions $[w_{uv}]_{i'j'} \leftarrow \lambda_{(vi'j')}[w_{uv}]_{i'j'}$ and $[w_{u'v}]_{i'j'} \leftarrow (1 - \lambda_{(vi'j')})[w_{uv}]_{i'j'}$ for some scaling vector $\lambda$, yielding:

$$\mathbf{E}\left[ \frac{\partial \mathcal{E}}{\partial [K_{pu}]_{ij}} \right] = (A\lambda)_{pij} \quad \text{and} \quad \mathbf{E}\left[ \frac{\partial \mathcal{E}}{\partial [K_{pu'}]_{ij}} \right] = (A(\mathbf{1} - \lambda))_{pij}. \tag{20}$$

The contribution of the two modified filters $u$ and $u'$ to the squared gradient norm obtained after cloning is again given by

$$\mathcal{H}(\lambda) = \lambda^T A^T A \lambda + (\mathbf{1} - \lambda)^T A^T A (\mathbf{1} - \lambda), \tag{21}$$

and the same analysis holds.

## B    EXPERIMENTAL SETUP

All our experiments are implemented in Tensorflow[3] and are run on a single consumer GPU, the memory size and processing speed of which speed are currently the bottleneck of the proposed method. When performing tunnel opening / weight reorganization, we accumulate the required gradients during a pass over the entire dataset in order to stay consistent with the theory. We have experimented with two approximations to this: First, one can compute the required gradients from only a partial pass through the dataset and second, one can use moving averages during the iterations before tunnel opening to accumulate an approximation to the required gradients. Since we are dealing with flat areas, it is reasonable to assume that these moving averages over recent iterates would give a good approximation to the quantities at the tunnel opening point. For our experiments, both approximations perform as well as performing a full dataset pass up to a certain degree of stochasticity and may be viable options in practice. However, even with stochastic approximations, the size of the networks that the model can be applied to is limited by computational power. Further work will go into developing computationally efficient implementations of our method.

We preprocess the used image datasets by subtracting the pixel means calculated over the entire dataset. During training, we randomize the order in which we proceed to the dataset after each epoch and we use a minibatch size of 256.

---

[3]Code to use our method will be made available upon publication.

Hyperparameters were chosen using grid search over the range of viable parameters and kept constant during comparisons within the same setup. Shared parameters, such as learning rate, were determined in a baseline run of the network (not stuck in flat areas), which also provided a baseline target loss to be reached. The choice of when to perform tunnel opening was done to select a point when sufficient steps without change of loss had been performed in order to ensure that the algorithm is stuck in the flat area. We have experimented with changing the opening point to later in the plateau and observed that the effect is the same: Optimization escapes the flat area after opening, irrespective of when exactly the opening happens.

As a technical detail, when performing our algorithm, we use a standard implementation of SVD for solving the required eigenvalue problem.

In the following, we describe the detailed setup for our individual experiments.

### B.1    ADDING FILTERS TO A SINGLE HIDDEN CONVOLUTIONAL LAYER

We build a simple network with a single hidden convolutional layer with a sigmoid nonlinearity, followed by a fully connected softmax layer. The convolutional layer contains 8 filters of size $5 \times 5$ in order to keep the network size small. We train the network on the MNIST dataset using SGD and a learning rate of $1.0$. For finding the initial flat area we use Newton's Method with the full empirical Hessian inverse and a step size of $0.1$ for 20 steps. When opening the tunnel, we add 2 filters to the initial 8, which we remove again when closing the tunnel. To keep our network structure simple, we do not employ tricks such as batch normalization, pooling or residual connections.

### B.2    PRACTICAL WEIGHT REORGANIZATION FOR A SINGLE HIDDEN CONVOLUTIONAL LAYER

The setup for this experiment is equivalent to the first one, except that instead of adding and removing filters, we perform our suggested practical weight reorganization.

### B.3    TUNNEL OPENING IN A DEEP NEURAL NETWORK

For the last experiment, we change our setup to a more realistic scenario. Our network consists of 5 hidden convolutional layer, each with 64 filters and ReLU nonlinearities. This is followed by a fully connected softmax layer with 1000 output units. We equip the first 3 layers with batch normalization and max pooling of stride 2. We train on a scaled version of the ImageNet dataset ($64 \times 64$ pixels) using Adam Kingma & Ba (2014) with a learning rate of $0.00002$. We find a flat area by exploring different random initializations until we get stuck in a large flat area. We then re-use this initialization for all experiments. However, a degree of stochasticity is provided by random shuffling of the dataset. This shows that first, the flat area is a property of the network, not of the exact data order and second, we get stuck (and are able to escape) in a reproducible manner. When performing opening, we add a single filter to each of the two last hidden layers. Immediately after opening, we clear Adam's accumulation buffers and reduce the learning rate by a factor of 10 for 20 steps as a warmup procedure for Adam. This is done both for the runs where we perform opening as well as during the control run. Still, there remains a small increase in the loss function after we apply our algorithm. Empirically, we observe that this increase vanishes when we use SGD after tunnel opening instead of Adam, which is evidence that the increase is due to the warm-restart effect of Adam adapting to the new network architecture.

## C    MORE EXPERIMENTAL RESULTS

### C.1    SYMMETRY BREAKING

We investigated the behavior of the network after applying our method with special regard to how the symmetry between the original filter $K_u$ and the copied filter $K_{u'}$ is broken. We measured the distance $\|K_u - K_{u'}\|_2$ experimentally for various settings of $\lambda^t$. According to our theory, using our proposed $\lambda^t$, this distance should increase very rapidly after tunnel opening, because the flat area is escaped quickly, while any other setting of $\lambda^t$ should lag behind significantly. In the special case where $\lambda^t$ is a vector entirely composed of $\frac{1}{2}$, the two units should be exactly the same and symmetry should never be broken.

Results can be seen in Figure 3. As predicted by our theory, using our proposed $\lambda^t$ escapes the flat area quickly and the two initially equal filters diverge rapidly during optimization. Compare this to the random (uniformly sampled) $\lambda^t$, which escapes only much later and more slowly, and thereby the symmetry between the filters is broken more slowly. Note that the iteration scale for this experiment is longer than the other plots, in order to show the final convergence of the filters to a common area of distance from each other, so the initial speed difference is significant. Further note that any fixed $\lambda^t$ to a vector of all same values will break symmetry slightly slower than random $\lambda^t$, with the clear excaption of $\lambda^t = \frac{1}{2}\mathbf{1}$, which never breaks symmetry and the distance remains at zero. This confirms our theoretical analysis.

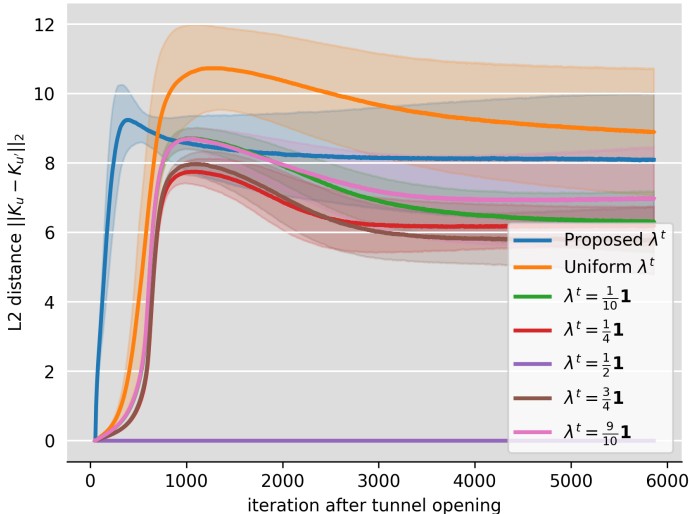

Figure 3: Distance between new filter $u'$ and filter $u$ from which it was copied during tunnel opening in MNIST experiment. Note that symmetry is broken faster using our proposed method, which is due to the network escaping the flat area more quickly. Uniformly sampled $\lambda^t$, as well as fixed $\lambda^t$ escape more slowly and therefore break symmetry more slowly. Note that in the special case $\lambda^t = \frac{1}{2}\mathbf{1}$, symmetry is never broken. Shaded areas are one standard deviation from the mean over 10 experiment repetitions.

## C.2   WALL CLOCK TIMES

In Table 3, we report the run times of our full experimental runs (as displayed in Figure 4). As can be seen, the overhead from the tunnel opening procedure is a one-time insignificant addition when compared to the variability of compute times.

## C.3   CLASSIFICATION ACCURACIES

Though our work investigates purely the escape from plateaus in training set loss, we measured the final classification accuracies of our obtained classifiers. Table 4 displays these. Note that we did not find a difference between networks that had been optimized using our method and networks that had not, except that the final accuracy is reached faster.

## C.4   TUNNEL OPENING AT LOCAL MINIMA

Since our theory is applicable at any point where the gradient vanishes, we could apply it not only in flat areas, but also in local optima (e.g. see Section 4.2). We investigated this using various pretrained architectures on the ImageNet dataset, including our own 5-layer CNN, but also variants of VGG (Simonyan & Zisserman, 2014).

| Dataset | Method | Wall clock time mean | (standard deviation) |
|---------|--------|---------------------|---------------------|
| MNIST | Control | 1420 | (218) |
| | Random | 1317 | (370) |
| | Proposed | 1318 | (284) |
| | Noisy SGD | 1390 | (236) |
| | NEON | 1367 | (251) |
| ImageNet | Control | 31222 | (2218) |
| | Random | 32812 | (2429) |
| | Proposed | 31361 | (2636) |
| | Noisy SGD | 31964 | (2927) |
| | NEON | 32518 | (2737) |

Table 3: Wall clock time (mean and standard deviation) of our experimental runs in seconds.

| Dataset | Layers | Filters per layer | Test set accuracy |
|---------|--------|-------------------|-------------------|
| MNIST | 1 | 8 | 0.98 |
| ImageNet | 5 | 64 | 0.28 |

Table 4: Classification accuracies of classifiers used in our experiments.

In accordance with our theory, we observed that tunnel opening leads to an increase in gradient norm, followed by a decrease in training loss, but we were unable to significantly increase the test set classification accuracies of the provided classifiers. More work remains to be done on the importance of switching between distinct local minima and their generalization properties.

## C.5 FULL EXPERIMENT RUNS

Figure 4 provides plots for the full runs of the experiments listed in the main section. Note that all optimization methods converge to the same loss (and the same test set accuracy).

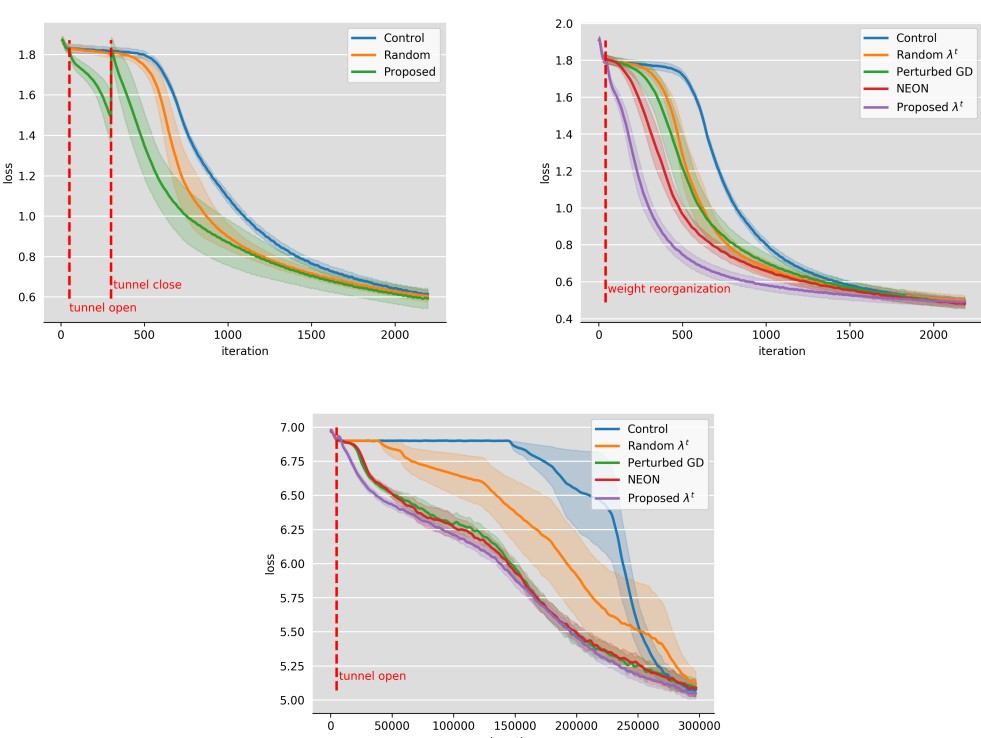

Figure 4: Full experiment runs. Top Left: 1-layer MNIST CNN opening and closing. Top Right: 1-layer MNIST CNN weight reorganization. Bottom: 5-layer ImageNet CNN tunnel opening.

