# OpenReview forum: "Escaping Flat Areas via Function-Preserving Structural Network Modifications"
_ICLR.cc/2019/Conference_

### Official Review · AnonReviewer2 · 2018-11-02

**Rating:** 6
**Confidence:** 4

**Review:**

The paper addresses the problem of increasing and decreasing the number of hidden nodes (aka, dimensionality) in the network such that the optimization will not enter the plateaus of saddle points. The opening or closing of tunnels (filters) guarantee the existence of “new escape directions” and faster convergence.

Strengths:
+ provide a new perspective of designing the shape/dimensionality of a network in a dynamic manner.
+ provide theoretic proof of CNNs and FCs on the contribution to the gradient after cloning.

Weakness:
- Experiments are very weak to verify the theory.

Detailed comments:

- Eqn. (6) seems to provide a unified evaluation on the contribution of two units to the gradients. How does it relate to the experiments? It gives me a sense that the manuscript is isolated between theory (Section 2 and 3) and verification (experiments).
- Why does the blue curve get stuck in a flat area? A better staring learning rate could alleviate the plateau bottleneck.
- The experiment settings are a little bit simple, even for the most complicated one in Section 4.4, where there are five conv layers and the tunnel opening only involves one single filter. Do authors conduct more filters opening in more layers? How about the closing case? There is no result/analysis in the experiments.
- Why authors claim the blue curve in the left figure 2, a “flat area”? It seems working as the orange one (loss decreases normally).
- Another big concern is that the proposed method is supposed to prevent network from saddle points and faster convergence, which is verified. And yet, the ultimate goal is to improve the performance. I am surprised that there is no such result at all in the manuscript (for example, error rate goes down on cifar/mnist/etc).

In summary, I do recognize the theoretical effort the paper has provided; however, the experiments seem not to verify the proposed method in a professional manner.

---

### Official Review · AnonReviewer1 · 2018-11-03
**Interesting ideas for expanding/contracting a network's size to escape saddles, but would benefit from further experiments**

**Rating:** 4
**Confidence:** 4

**Review:**

Summary:

This paper presents a new strategy for escaping saddle points by adding and removing hidden units during training. The method essentially finds conditions where adding a hidden unit does not change the overall input-output map, and uses these as constraints to add a hidden unit that maximally increases the gradient norm (thus potentially getting learning unstuck). Experiments show that the method can improve training speed relative to the same network with randomly added new units.

Major comments:

This paper presents interesting theoretical ideas and clearly separates the opportunities for adding/removing hidden units without changing the input-output map from the impact on the gradient due to the change in parametrization.

The experiments show that the proposed method can speed up learning when a network is genuinely stuck at a saddle point. Importantly however, the experimental evaluation intentionally seeks saddle points using Newton’s method, such that learning is genuinely stuck, before adding the additional units. It is therefore less clear whether this method can offer speedups to network training in practice. Do NNs come close enough to saddle points to benefit from the method when beginning from typical initializations? Experiments on ImageNet begin from a specifically chosen random seed that happens to enter a very flat region. How many random seeds were tried before finding this one? This would speak to the importance of these findings in general. The paper would benefit greatly from applying the proposed method to networks trained under standard protocols, to identify the speed up (if any) it can confer for the average case. It would also be important to account for wall clock time, as the proposed method involves potentially expensive steps (at least in its straightforward form).

The paper notes several other strategies for expelling from saddle points. The experimental evaluation could be improved greatly by including comparisons to these alternatives. Does the proposed method escape more quickly, or have other merits relative to these alternatives?

The clarity of the paper is good overall but the title could be improved to be more informative of the content of the method.

Overall the significance of the paper is not clearly established because the evaluations mostly consist of internal comparisons, in somewhat unnatural settings (where networks are initialized right next to saddle points). The theoretical observations, however, seem promising.

Minor comments:

I could not understand the motivation for closing the tunnel—it seems as though optimization proceeds more quickly if it remains open.

The paper discusses a range of relevant work but would benefit from citing other incremental learning work in neural networks, in particular:

Y. Bengio et al., Convex Neural Networks, NIPS 2006

F. Bach. Breaking the curse of dimensionality with convex neural networks. JMLR 2017

---

> ### Author Response · Authors · 2018-11-22
> **Thank You**
>
> Thank you very much for your constructive comments. We address them in order as they appear in the feedback.
>
> First, we recognize that using Newton's method and random restarts (we did ~150 and chose the "flattest" one) in order to explicitly find saddle points puts us in a constructed scenario and is somewhat dissociated from actual practice. However, being stuck in a flat area is a pre-requirement for our analysis and thus, since our experiments should serve to support our theory, we chose to match the requirements as close as possible. Almost all published research on escaping saddle points does pose being stuck in a saddle point as e pre-requirement.
> Second, we are aware that there is an ongoing, unresolved discussion in the community about how important the problem of saddle points is in deep learning. Our goal is not to add to this discussion, but to present a solution in one possible case.
> That said, we do report in the appendix what happens if the method is applied to models trained under a standard protocol (nothing happens, because they seem to reach minima instead of saddles), and we have added a table to the appendix reporting the wall clock times of all experiments. From these, one can see that the time overhead caused by our method is insignificant compared to the time variance over experimental repetitions (the computational bottleneck of our method is dominated by memory).
>
> As for your comments on comparison to other methods, we agree and thus we have implemented two baseline methods and included them in the experimental sections. Note that while these baselines require changing the gradient update over multiple iterations, our method is done within a single step and still outperforms them in terms of how fast the flat area is escaped from.
>
> We changed the title to be more informative. We've also removed the word "magic" from our text in order to make it more concise.
>
> Your observation that closing the tunnel does hurt the optimization procedure is correct and does not have to be done in practice. Our motivation for closing the tunnel is twofold: First, we wanted to establish a method that does not change the final size of the network. Second, closing the tunnel is a half-way step to our practical weight reorganization method, which does not change the network architecture at all and therefore might be much more implementable in current deep learning libraries.
>
> Lastly, thank you for the relevant related work. We have included citations to the papers.
>
> Besides the changes resulting from your feedback, we have made several other changes, motivated by the other reviewers. Most notable are the change in notation of network weights in order to better conform to the community's practices, the introduction of experiments on symmetry breaking in the appendix and the drop of the additional experiments on CIFAR10 from the appendix, since the network was arguably not stuck in a true flat area.

---

### Official Review · AnonReviewer3 · 2018-11-04
**Interesting work, though not sure how symmetry breaking happens**

**Rating:** 6
**Confidence:** 3

**Review:**

This is an interesting submission and I appreciate especially connecting to a body of literature which is not normally well known in our community (e.g. Fukumizu&Amari). I think the perspective is definitely new and probably quite relevant not only for practical approaches to escape saddles but also to understand learning in deep learning.  I have a few notes and suggestions:

1) Name of the paper:
I think is not descriptive of the approach and actually the words “magic” makes it sound strange. I think this will reduce the amount of people reading the work. Please consider something more descriptive like: “Escaping saddle points by increasing capacity”. Or something else more inspired, but that also hints what the work is about.

2) Notation:
The notation used is not ML friendly (or generically) to the average reader. I strongly suggest to use b_v for bias, W_uv for weights, and not theta_v and theta_uv which is not typical notation. ‘u’ and ‘v’ are somewhat non-typical choices either, though I understand that they come from the graph notation. Transfer functions are usual sigma. In the text you explain the process by starting with a u’ and then add the clone which is u. Normally you should have started with u and add the clone that is u’. x_u for the value of unit u (assuming this is in the middle of a deep net) is also quite a strange notation. I can guess the authors might be from a slightly different community, but I’m worried about people from the target audience (ICLR) being turned away from the work or even worse confused because of notations.

3) Related work
There is the Net2Net work that is related to what is going on here that is not cited (https://arxiv.org/pdf/1511.05641.pdf). I think there was some follow-up work after this.

4) Symmetry breaking
I do not understand how symmetry is broken. If I clone a unit, and have a new variant of it u’ that now has the same outbound connections but multiplied by alpha (while the original unit by 1-alpha) then while the norm of the gradients differ, their direction does not. Wouldn’t this mean that the units will track each other and hence no tunnel is open? In Net2Net dropout was used to break symmetry (i.e. a source of noise that would pick one path over the other). There is no source of noise here to break symmetry.

5) Diagrams and analysis
Connected to this, I feel like this could have been represented clearly with a diagram showing the net before and after. There could be some analysis, a more extended discussion of where the symmetry breaking comes from, empirical evidence that it does. I’m not necessarily worried that experiments are not scaled up, I’m more concerned that the hypothesis and solution is only tested by means of change in performance. What is this tunnel doing? How does it change the Hessian at the saddle? Any visualization to reinforce the intuition of what the approach is doing?


6) Closing tunnels & re-organizing
I don’t understand the mechanism for reorganizing weights and closing tunnels. It seems first of all to confirm my intuition that there is no symmetry breaking since we can “close” the tunnel by simple algebraic manipulation. So if those two units always stay in sync how do you actually change the error surface? How do you take advantage of this extra capacity to solve anything. Regardless, when it comes to re-organizing, it seems you pick two of these units that are in sync (previous tunnel I guess) and collapse them to open a new tunnel, right? How does this change anything? Which unit needs to be cloned? Any? So then why is the previous tunnel not efficient anymore?

---

> ### Author Response · Authors · 2018-11-22
> **Thank You**
>
> Thank you for these extensive and helpful comments. We'll address them in order:
>
> 1.) Name of the Paper
> We agree and we have changed the paper's name to a more descriptive one. Also we've dropped the word "Magic" from the paper to achieve more clarity.
>
> 2.) Notation
> We agree that the notation stood to be improved and we've changed all thetas to "w" and "b" for weights and biases. Further, we've swapped "u" and "u'", where now u refers to the original node to be cloned from and u' refers to the newly created, copied node. We believe these changes should make the work more understandable for the community.
>
> 3.) Related Work
> We've changed our related work section to include Net2Net and similar work and describe the main difference: This work is done in the context of transfer learning and architecture search, not in the context of escaping flat areas. While these works also do preserve the network's input-output function, our main contribution is that we provide an explicit way to control the gradient after the architecture change. This can be seen most evidently in hidden unit cloning, where these works clone units with weights of 0.5 equally (which leads to remaining stuck in a flat area as well as symmetry problems), where as we use \lambda^t with direct relation to the resulting gradient.
>
> 4.) Symmetry breaking
> As said before, it is true that when nodes are cloned with weights 0.5 equally, then the resulting network will have a symmetry that cannot be broken. Net2Net solves this by introducing noise (losing theoretical guarantees). However, this is only the case when the shares are exactly 0.5, where as we use weights of \lambda and (1-\lambda) where lambda is not only a number, but an entire vector. It's true that at first this just results in differently scaled, but otherwise same, activations from both units, but through the nonlinearities in the network, even this directional symmetry will be broken. Over multiple steps, the units diverge from each other.
> In order to show this, we've performed a series of experiments and made a section in the appendix (called "symmetry breaking") where we show the results: We've observed the L2 distance between the two units u and u' over the number of steps after we perform tunnel opening. What is visible is that only when \lambda is exactly 0.5 everywhere, there is a symmetry that can't be broken. Any other \lambda, be this another fixed number, a randomly sampled vector or our proposed \lambda^t will lead to symmetry being broken over time. Note that the experimental results suggest that using our proposed method, symmetry is broken significantly faster than using any other method. We hope this answers your question.
>
> 5.) Diagrams and analysis
> We would have loved to include visual diagrams of the network before and after, but ultimately had to prioritize other things for the limited space. In terms of analysis, we hope that the newly added experiments on symmetry breaking serve this purpose.
>
> 6.) Closing tunnels & re-organizing
> Closing the tunnel without effect can only be done when there have been no gradient update steps since opening, as we state in the paper. When gradient update steps are performed, the cloned nodes get out of sync and that's how the extra capacity is used. This also means that when closing the tunnel after a number of steps, it becomes a heuristic, i.e. we cannot guarantee that the input-output function is preserved. In fact, it almost certainly isn't. We can only give such a guarantee for opening the tunnel. Our goal was to make a method that doesn't change the final capacity of the network and this heuristic is a half-way step towards our other heuristic of practical re-organization, which does not change the architecture at all. We have amended the section on tunnel closing to hopefully clarify this further.
>
> Besides the changes resulting from your feedback, we have made several other changes, motivated by the other reviewers. Most notable are the addition of two baseline methods as well as the drop of the additional experiments on CIFAR10 from the appendix, since the network was arguably not stuck in a true flat area.

---

### Meta-Review · Area_Chair1 · 2018-12-13

**Confidence:** 3
**Recommendation:** Reject

**Metareview:**

The paper proposes a method to escape saddle points by adding and removing units during training. The method does so by preserving the function when the unit is added while increasing the gradient norm to move away from the critical point. The experimental evaluation shows that the proposed method does escape when positioned at a saddle point - as found by the Newton method. The reviewers find the theoretical ideas interesting and novel, but they raised concerns about the method's applicability for typical initializations, the experimental setup, as well as the terminology used in the paper. The title and terminology were improved with the revision, but the other issues were not sufficiently addressed.